# Learning Circles: A Collaborative Approach to Enhance Local, Healthy and Traditional Foods for Youth in the Northerly Community of Hazelton/Upper Skeena, British Columbia, Canada

**DOI:** 10.3390/ijerph192315878

**Published:** 2022-11-29

**Authors:** Louise W. McEachern, Sandra Harris, Renata Valaitis, Anissa Watson, Jennifer Yessis, Barbara Zupko, Rhona M. Hanning

**Affiliations:** 1School of Public Health Sciences, University of Waterloo, Waterloo, ON N2L 3G1, Canada; 2Gitxsan Government Commission, Hazelton, BC V0J 2A0, Canada; 3Storytellers Foundation, Hazelton, BC V0J 2A0, Canada

**Keywords:** indigenous health, food sovereignty, food skills program, school food, traditional food

## Abstract

Youth health, long-term food sovereignty and the reclamation of traditional food-related knowledge are areas of concern within Indigenous communities in Canada. Learning Circles: Local Healthy Food to School (LC:LHF2S) built on an exemplar program in four predominantly Indigenous communities. In each, the initiative worked with interested community members to plan, implement and evaluate a range of activities aimed at enhancing access to local, healthy and traditional foods for schools and youth. This case study describes the context, process, outcomes and perceptions of implementation in one of the communities, Hazelton/Upper Skeena, located in northern British Columbia. Data were collected between 2016–2019 and included semi-directed interviews with community members and LCEF (n = 18), process reporting (e.g., LCEF reports, emails, conference calls and tracking data), photographs and video footage, and photovoice. Data were analyzed thematically. Hazelton/Upper Skeena has an active local and traditional food culture. Indigenous governance was supportive, and community members focused on partnership and leadership development, gardens, and food skills work. Findings point to strengths; traditional food, knowledge and practices are valued by youth and were prioritized. LC:LHF2S is a flexible initiative that aims to engage the broader community, and exemplifies some of the best practices recommended for community-based initiatives within Indigenous communities. Results indicate that a LC is a feasible venture in this community; one that can facilitate partnership-building and contribute to increased access to local and traditional food among school-aged youth. Recommendations based on community input may help the uptake of the model in similar communities across Canada, and globally.

## 1. Introduction

Youth health, long-term food sovereignty and the reclamation of traditional food-related knowledge are areas of concern within Indigenous communities in Canada. Consequences of colonization for Indigenous peoples include loss of traditional food systems; additionally, prevalence rates for chronic disease for all age ranges including adolescents remain higher than those among the non-Indigenous population [1,2]. Problems such as lifestyle-related chronic disease are complicated, and programs to address this call for many and varied system components to work together coherently to see the desired benefit [3]. Strong partnerships, and clear and honest communication between all stakeholders are characteristic of successful programs that attempt to address complex problems [3,4].

‘Health’, and health promotion for Indigenous communities is based on the concept of wellness, a balance between physical, spiritual, emotional and mental health [5,6,7,8] in contrast to the traditional focus of Western science on physical health. However, all too frequently in the past, research and related community health programs involving Indigenous people were conducted from the perspective of Western scientific thought, with little attention paid to the views the communities involved. Indigenous communities have felt like objects of scrutiny and interest rather than treated respectfully as community partners and end-users of the study results or program goals [9]. Many communities were prevented from accessing the data resulting from research in their communities, and in some cases, study results were not shared [8,10,11]. A return to a culture of healthy local foods and greater food sovereignty is fundamental to the promotion of health in Indigenous and remote school communities [7,12].

When working with Indigenous communities to conduct health research, community-based participatory research that emphasizes the value of partnership creation between researchers and community members has been shown to be effective. This is particularly true when enabling sustainable change [13,14]. Additionally, the principles of ownership, control, access and possession, or OCAP, have been developed as a response to such unsuitable research practices, and apply to all aspects of the research process with First Nation communities. When navigating the process of establishing shared ownership over group data, these principles can be a helpful guide to both researchers and community members [11]. Using OCAP as a guide, the goal of the Learning Circles: Local Healthy Food to School (LC:LHF2S) initiative has been to collaboratively enhance holistic health within partnering remote Indigenous school communities.

The Learning Circle (LC) approach is a novel community engagement practice, working at individual and community levels, with the overarching goal to promote partnerships between community members with a common interest in food. The model first began as the ‘learning labs’ of the Farm to School programs in the United States and transitioned to Canada under Farm to Cafeteria Canada in 2013 (F2CC) [15]. The purpose of this flexible initiative was to increase the provision and inclusion of local healthy food in schools. Encouraging results from the United States, and the Canadian pilot community, Haida Gwaii, B.C. [16,17], motivated the wider project team to scale the approach up by including three new communities in 2016 and supporting ongoing work in Haida Gwaii. In contrast to previous programs, each LC is adapted according to the wishes of the community—therefore the model was thought to be appropriate to accommodate the diversity in context between communities [18,19]. Little was known at the outset of the project about scale-up in remote First Nation (FN) communities and remote school communities [20,21], although the goal of the scale-up was to encourage sustainable change and increase the impact and reach of the approach [22] in each new community.

This specific paper describes the food, nutrition, and health related transitions (context, process, activities, influences, outcomes) of the LC in the community of Hazelton/Upper Skeena (HZ), British Columbia, Canada, from January 2016–December 2018. This case study also evaluates perceptions of the transition in local food systems, partnership development, food literacy among youth, and community capacity associated with the LC in this community.

## 2. Methods

### 2.1. Study Design and Area

This study was conducted using community-based participatory research (CBPR) [23] principles and collected data using various methods (Table 1) selected by community participants to help identify program strengths and facilitators [23,24]. Community members had agency and control over their own data, its collection and interpretation [25,26]. The case description was co-developed by community advisors, community partners and research team members to center community context, activities and voice.

### 2.2. The Learning Circle

Details of the development of the LC community engagement practice, and role of the Learning Circle Evaluation Facilitator (LCEF) are described elsewhere [17]. For the LC: LHF2S initiative discussed here, partnerships were generated around a shared interest in increasing the availability, acceptability and consumption of local, healthy and traditional foods by school-aged youth and adolescents in each community. Partnerships with key community organizations were developed and community members with an interest in local food were invited through formal and informal networks. Priorities for that community were agreed upon over the course of a workshop (usually day-long), needs and activities were discussed, and goals set. The core group met between 2016–2019 to re-evaluate the aims of the group and discuss progress.

A critical component of the research process of the larger project was Annual Gatherings where representatives from each community, including the LCEF gathered with diverse stakeholders (researchers, partners, and members of other participating communities) annually to share knowledge, experiences, and provide support in the research process [27]

### 2.3. Ethics

Ethical clearance was received from the Office of Research Ethics at the University of Waterloo, as part of the evaluation activities for the LC:LHF2S study (ORE# 30819). Conduct of evaluation activities was established between the University of Waterloo and the community of HZ in an informal way. Centering decisions regarding what information was desired, and which processes were used for collecting information, locally reflects how OCAP and principles of respect and reciprocity were honored in this research.

### 2.4. Data Sources and Data Collection

Evaluation methods were guided by community priorities and multiple types and sources of data were analyzed to describe the story of change resulting from the work of the LCs in HZ. 

Interviews: Annual interviews were conducted with LCEFs, key community members and project partners using a semi-structured interview script. All participants were closely connected with the community and the program; sociodemographic information was not collected. Participants were purposively selected based on their connection with the initiative (for example, involvement with the supporting organization, Storytellers’ Foundation or the Gitksan Government Commission), and were recruited by email or in person. Interviews took place in person, by phone, and on Skype, lasted between 25–60 min, and were carried out, in English, by the same interviewer (LWM). End of project interviews with community members were conducted by a trained research assistant chosen by the community. All interviewees provided permission for the interview to be audio-recorded and signed and/or provided verbal consent for use of the data in future publications. Participants were not compensated, had the opportunity to withdraw from the interview at any time, and could decline to answer any question asked during the interview. 

Written documentation: Notes taken during conference calls between project partners that took place through the duration of the LHF2S initiative, and emails exchanged throughout the project were analyzed as written documentation. After each LC, the LCEF wrote a report describing the participants, events and action plans generated by the meeting. Activity tracking data, timesheets, and journals from the LCEF, teleconference minutes and descriptions of workshop/food skills class were collected throughout the initiative and emailed to the research team at the University of Waterloo. These documents also contributed to annual reports to the funder (Canadian Institutes of Health Research (CIHR)) and partners. 

Photographs: Digital photographs and videos were taken by LCEFs, community members, and the research team throughout the initiative. Additionally, a photo-voice project was carried out with school students. Students were instructed to take photographs of things they thought were important to telling their story about the foods that are available in their community, using the following points as their guide:Where: Places that affect what you eatWho: People that affect what you eatWhy: Information that helps you decide what to eatOther: like price of foods you might choose

Students were instructed to take as many photographs as they liked, and to select 15 for discussion in class. They were also given guidelines regarding appropriate behavior in obtaining photographs.

### 2.5. Data Analysis

Audio-files from all interviewers were transcribed verbatim and pseudonyms were assigned to each participant immediately after data collection to protect anonymity. Transcripts, field notes, reports, and all other written documentation, including photographs, were reviewed and coded using the following steps. Activity tracking documents and early participant interviews were used to develop a draft coding framework. Both deductive approaches (exploring the data for themes relating to wider project research objectives) and inductive approaches (observing themes emerging from the data) were used in the development of the codebook, in discussion with the project research team. Data were then coded line-by-line (reports of LCEF activities were coded and summarized by one coder (BZ), and coding of interviews was carried out in duplicate (LWM and RV)) and analyzed for emerging themes by comparison across the documentation. A selection of interview transcripts was returned to the community for member checking, to support methodological rigor; one interview transcript was withdrawn from the analysis after this step. Data processing and analysis was carried out using NVivo^®^ 12 Pro (QSR International). The large variety of data sources included in this study allowed for triangulation of data, contributing to methodological rigor [28,29,30]. 

Main themes were discussed with the communities remotely, due to the COVID-19 pandemic, as part of sense-making of the themes that emerged during analysis. 

### 2.6. Context 

HZ is located in the upper Skeena River region in Northwest British Columbia, approximately 300 km inland from Prince Rupert. The region is located on the traditional territories of the Gitksan First Nations (Alternative spelling: Gitxsan First Nations) and encompasses 14 distinct communities [31]. About 5000 people live in two municipalities: Hazelton, and New Hazelton; seven reserves; two nonincorporated settlements, and three valleys. The Gitksan First Nation makes up almost 80% of the population with most of the remainder being of Western European descent; there is some representation also from the Wet’suwet’en Nation in the region [31]. The land is mountainous, covered with spruce, subalpine fir, hemlock, cedar and pine forests, and fed by the salmon-rich Skeena River. Foods traditionally eaten in this region include salmon, oolichan, wild meats (such as moose), berries, vegetables such as root vegetables, and traditional medicinal plants. 

The LC in this community received administrative support from Storytellers’ Foundation, a non-governmental organization based in the community since 1994, and visionary direction and advisory support from the Gitksan Government Commission (GGC)—a form of tribal council to support four of the six Gitksan Band Councils in Upper Skeena. 

Some food-related initiatives (i.e., school-based food programs, food security work and youth-based land programming) had been taking place in the region prior to the beginning of the LC Initiative in 2016. A Wellbeing Model had been developed by the GGC that aimed to depict the balance between personal, political, social and spiritual aspects of life links healthy people, healthy communities, and healthy lands [32]. This model included a photovoice mural project and aimed to provide a way for community members to engage with Gitksan culture. Chronic illness and health, along with food insecurity, had been identified by the community in a Community Planning process prior to 2016, as key issues that needed to be addressed.

Poverty and other systemic barriers to health were identified in interviews as general challenges that affect the communities in this region. In a survey, separate from the current project, conducted among students in the school district (SD #82) in 2018, 21% of students (n ≥ 500, age 12 to 18 years) reported never eating breakfast on school days with 6% never eating lunch; 35% reported sometimes eating breakfast, 35% sometimes eating lunch and 11% sometimes eating dinner; and 10% reported sometimes/often/always going to bed hungry due to insufficient money for food at home [33]. The same survey asked students which of the following foods they had eaten on the previous day. Foods reported as eaten ‘once or twice’ were fruit (59%); vegetables or salad (62%); traditional foods (19%); food grown/caught by the student or their family (18%); sweets (60%) and fast food (43%) [33].

The Upper Skeena River region is part of the B.C. School District #82 and has eight schools: one secondary school and seven elementary schools. Four of the elementary schools are located on-reserve and are independent schools, administered by the separate Bands. Of these schools, three (HSS, MGA, NHES) became involved in the LC initiative.

## 3. Results

### 3.1. The Learning Circle Support and Staffing 

The LC in this community received administrative and advisory support from the Storytellers’ Foundation; the organization was responsible for advertising for and hiring the LCEF. A service agreement in 2016 with Storytellers’ enabled the University of Waterloo to send funds directly to the community for certain project related costs including the salary of the LCEF and honoraria, e.g., for Elders. The LCEF position in HZ was advertised in the community and appointed by Storytellers’ Foundation with input from representatives from the GGC and began work in the Fall of 2016. This LCEF went on leave in 2017 and was replaced by another for the duration of the project funding. Five LCs were held in HZ between Feb 2017 and Nov 2018.

The consistent attendees of the LC in HZ included representatives of local Indigenous governance (GGC, Band Council), staff members from four local schools (one nursery school: Gitanmaax Nursery School; two elementary: Majagaleehl Gali Aks School (MGA) and New Hazleton Elementary (NHES); one high school: Hazelton Secondary School (HSS)); locally based NGOs (Storytellers’ Foundation); Skeena Watershed Conservation Coalition; Senden Sustainable Agricultural Resource Centre (Senden); North West Food Action Network) and local farms (Wood Grain Farm; Dancing Bee Farm).

The second Annual Gathering of the full LC:LHF2S initiative was held in HZ, close to the community of Smithers. A visit to the various communities was arranged and some cultural activities for the group to experience were facilitated, such as local singing and drumming, and a tour of Senden and market gardens. The pre-existing program at Senden focused on gardening and other food skills with local youth (see Table 2).

### 3.2. Case Sample and Source Documents

Between 2016 and 2018, 12 informants gave 18 interviews; of the 12 participants, two acted in a facilitatory role and two acted in advisory role as related to the LC. Eight were community members. An LC report was written after each LC leading to 5 in total, 13 activity tracking reports were submitted by the LCEF, and the minutes of 15 meetings between the LCEF and the University of Waterloo research team were analyzed. A photovoice project was conducted among some of the youth in 2017 with 15 entries submitted, and 9 photographs taken in and by the community were available for analysis (Table 1).

### 3.3. Goal Definition, Partnerships and Funding

The LC in HZ decided upon five main areas of focus: Funding: Seek funding to augment current food-related activities within the communityLocal food: Value local food and work to increase its value among community membersFood security and skills: Purchase more local food and provide growers, harvesters, and school staff with the tools to make healthy food options viableGardens: Aim for every school to have a garden and a greenhouse to get kids outside on curriculum-based projects that broaden students’ knowledge, experience and skill with local foodYouth: Connect kids with a healthy food culture on the territories.

Partners of the LC in HZ, and their associated activities, are listed in detail in Table 2.

At each LC these goals were revisited and progress towards them assessed. By the end of the funding period, 5 grants (F2CC × 2, Northern Health, Mazon Canada, Whole Kids Foundation), were applied for by three schools and 4 received. These grants were used for building and expanding gardens and greenhouses, purchasing tower gardens and a vermicomposter for use in school, and local food procurement for school food programs. A further $6000 was raised at a community-based Mid-winter gala in 2018, with the funds allocated for schools to bring students out onto territories for food-related activities adhering to traditional protocol.

Three of the eight schools in the area had their own gardens/greenhouse by 2018.

### 3.4. Activities

The Gitksan Wellbeing model provided a baseline focus for the LC meetings. Activities of the LC focused on partnership building; gardening, developing food skills, and connecting youth to traditional food activities and the land. Each of these types of activities are described in this section and in Table 2. The LC was also used to showcase local products and foods.

#### 3.4.1. Activities: Partnership Development 

Partnership development as a function of the LC was focused on, particularly at the beginning, but continued throughout. Connections were made between the LCEF and various community stakeholders such as Gitksan Health, First Nations Health Authority (FNHA), Northern Health, Skeena Watershed Conservation Coalition, schools, and Senden. Relationships were developed with farmers in order to procure local foods for school food programs and community events.

These connections were used to put on a number of community-wide events, such as the Wellness Food Festival (partnership between Storytellers’ and GGC) in the winter of 2017/2018. A community-based Mid-Winter Gala was held in January of 2018 and featured local foods donated by farmers and prepared by local chefs, students as food service staff and vendors of their artwork; this gala raised over $6000 and was instrumental in increasing the visibility of the work of the LC. A weekly ‘Intergenerational Community Kitchen’ program that aimed to build food skills and literacy between Elders and youth was begun in the Autumn of 2018, as well as a 6-week ‘Kids Get Food’ course that took place at Storytellers’ Foundation. Partnerships between schools and Senden resulted in visits from staff and students to Senden for community days and skill-development (Table 2). A mentorship program was set up between a secondary school (HSS) and elementary school (MGA), with elementary students visiting the HSS garden and root cellar.

Connections were also built with other communities involved in the project, particularly the LC in Haida Gwaii. Students from HSS visited Haida Gwaii in the Spring of 2018 to exchange knowledge, food, and medicines.

#### 3.4.2. Activities: Gardens

Two elementary schools developed gardening programs for their students. Children attending the Gitanmaax Nursery School became involved in the local community garden in Gitanmaax, learning to tend the flower and vegetable portions of the garden. At MGA school, garden boxes were built, and funding obtained for a greenhouse. Garden beds were built at NHES (school). Funding was obtained by HSS (school) to purchase tower gardens, which were used to grow traditional foods, and a vermicomposter was purchased (Figure 1). Work continued at a root cellar and a smokehouse that had been built at HSS prior to the LC funding.

The school gardens were used as a mental health support for some students—a place to go to calm themselves or work out problems—and a memory garden was planned for HSS and MGA schools to assist with grief and be used as a place where students could go to be quiet in remembrance of family members that have passed away.

The Skeena Watershed Conservation Coalition was involved in the continuance of community gardens in Sik-e-dakh and Gitanyow communities, and a focus of activity at HSS was the ‘Back to the Land’ program for youth, in which they learned to grow vegetables for a Community-Supported Agriculture box program. Storytellers’ organized a Back-Yard Gardening program that met once a week and focused on assisting with community and school gardens in the region.

#### 3.4.3. Activities: Building Knowledge and Skills

A number of programs focusing on skills development were built on or developed during the course of the LC initiative. At HSS, the ‘Back to the Land’ cultural program focused on developing traditional knowledge and skills through experiential learning, for example hunting, gardening, maintaining the root cellar and smokehouse. Students involved in this program visited Haida Gwaii in the Spring of 2018 for knowledge exchange. Foods classes went to Senden once a week, and workshops were organized with local Elders to learn about traditional foods. Students could participate in filleting/butchering lessons for fish and moose. Seed planting workshops and field trips to local farms and greenhouses were organized by NHES, and chicks were incubated and hatched in the school in 2017. MGA initiated food skills classes (for example soup-making, dehydration and food preservation) and developed the gardening skills of their students in the school and community gardens.

At Senden, youth attending the program participated in traditional food processing workshops, for example fish smoking and canning, as well as medicine workshops with the Elders. Beehives at Senden allowed youth to develop bee-keeping skills. The Skeena Watershed Conservation Coalition ran skills courses that aimed to connect youth to the Skeena River and to their culture, and the Northwest Food Action Network ran a gardening skills conference called ‘Better Together’ in 2018.

A weekly Intergenerational Community Kitchen program that built food skills and literacy in Elders, youth, adult, and families had begun prior to the LC project, and work continued from 2016–2019.

In addition, research capacity was enhanced, for example, a community advisor attended CIHR’s annual gathering of all Pathways-supported projects; a research assistant was hired and trained to conduct community interviews, and the LCEF and one community member attended and presented at each of four ‘annual’ gatherings. The HZ community hosted the annual gathering of 2016.

#### 3.4.4. Activities: Connecting Youth to the Land and to Traditional Food Activities

The GGC ‘Brighter Futures’ program organized trips for youth onto territorial land for berry and medicine picking. The ‘Youth on Water’ river rafting program run by the Skeena Watershed Conservation Coalition focused on connecting youth to their culture, and local food was included in this program, facilitated by Storytellers’. The land-based education for youth at Senden focused on culture and traditional ways and as part of this the Gitxsenimx language was incorporated into all aspects of the program. 

At HSS (school) the ‘Back to the Land’ cultural trips aimed to connect youth to the land and traditional activities, and the school instituted an ‘Indigenous People’s Celebration Day’ in 2018. Each school was encouraged to invite Elders and traditional knowledge holders into the classrooms and teach children the Gitksan words for food and plants in the garden. 

Programs in HZ were developed with the thought of encouraging youth to connect with the community. Students became involved in picking extra apples from the community trees and made apple sauce which was brought as gifts to Elders. Students were brought to community events celebrating local or traditional foods, for example the Kispiox Salmon Bake, Senden Community Days, and All Clans Feasts.

#### 3.4.5. Activities: Food Security

Several organizations had been involved in strengthening food security in the community prior to the LC initiative beginning; the LC initiative strengthened food security by building connections between these organizations and providing time to build a coordinated approach. Food security was a focus for the Skeena Watershed Conservation Coalition who worked with the communities of Gitanyow and Gitwangak. The Gitwangak Food Security Partnership was formed, where $8000 of organic produce was delivered to the community of Gitwangak. The GGC organises Community Christmas Hampers filled with frozen or preserved produce from community gardens, and Storytellers’ ran a ‘Kids Get Food’ program that taught school children nutrition and food skills. Provision of school food programs (e.g., the salad bar) at MGA aimed to help address food security challenges among students attending the school.

### 3.5. Lessons Learned

Four main themes relating to the story of the local foods emerged from the data in HZ: Traditional Food, Knowledge and Practice; Building Partnerships; Learning from One Another; and Local Food.

#### 3.5.1. Traditional Food, Knowledge and Practice

A major theme that emerged from the data was that of the importance of culture and tradition as a bedrock for the LC work in HZ. The first LC introduced the Gitksan Wellbeing Model, newly developed by the GGC, that illustrates the interconnectedness of land/knowledge, spirit and learning and aims to connects young people to wellbeing through relationship with their culture, family, food and land [28]. 

*“…having the teachers involved, and … just seeing the little ones excited about good healthy food—and the high school students, and just taking pride in Gitksan culture, and learning, or connecting to Elders that have such a rich understanding of their homelands, and I think that’s a really, really important aspect of learning that we often don’t see in the school, so I just think that’s so valuable.”* (LC Participant and GGC representative)

One of the LC goals focused on this (‘Connect kids with a healthy food culture on the territories’) and the role that traditional food, skills and knowledge plays in food security was also acknowledged. ‘*That strengthens our lives!*’ is a translation of a slogan displayed in HSS: “*ent si daxgyat ga gandidilst’m*”, referring not only to healthy food, but traditional culture and the importance of the land (Figure 1). 

*“The youth were very proud of their Three Sisters-corn, squash and beans planted together are beneficial to each other when growing.’ Each youth had their own patch at Senden that they cared for and tended”* (LC Report)

*“…it is important our people realise that our youth are going back to gardening and we as Elders need to show them, that yes, we can put a garden in. Yes, we can do it!”* (Community Elder)

The inherent sustainability of traditional ways was discussed in a number of sources. In addition, the language used for and around food was considered important, and traditional protocols were carefully considered. 

*“Comparing jarred salmon [with purchased canned salmon]…highlighting the value people have for local and traditional foods”* (Photovoice entry, accompanied by a photograph of salmon jarred using traditional methods)

*“School gardens built and maintained over the course of the project-building the infrastructure at the school and leaving a legacy of the project”* (LCEF Report)

There was a tension between the concept of ‘local foods’ and that of ‘traditional foods’, with ‘traditional’ foods being the preferred term for Indigenous community members. Engaging the region’s predominantly Indigenous schools was found to be difficult, due, in part, to the LCEF being a relatively new non-Gitksan community member and not having the time to build necessary relationships, and in general, challenges relating to colonization, i.e., the school system, agriculture and gardening were part of this theme. 

*“Our whole relations with agriculture, with farming, has obviously been a strained one since [colonial] contact. You know our people have been displaced from the land by farmers, and you know the government giving away land or selling land. So, you know our people have been displaced. And so, there is a bit of a strain there. a bit of historical relations piece that needs some work or attention. That could be through reconciliation, that could be through you know working together.”* (LC Participant and GGC representative)

However, connection with the land using traditional ways is seen as important to community members as it brings healing from past traumas. 

*“The land offers us medicine and healing, when we harvest from the land we are healing ourselves.”* (Photovoice entry, accompanied by a photograph of drawers containing dried plants and herbs for medicines)

Systemic barriers related to food safety and legislative issues of serving traditional foods in schools, plus logistical issues related to school programs were noted. They lack a processing and storage facility in the community for local and wild game. Community members were interested to learn of the processing and storage facility in Haida Gwaii that enables safely bringing traditional food like moose and venison into schools for consumption by students.

#### 3.5.2. Building Partnerships

HZ had some local and traditional food work happening prior to the establishment of the LC and building relationships between interested stakeholders was a high priority. Connections resulted in community events such as the mid-winter gala which brought a large number of community members together in support of the initiative. 

*“I think one of the biggest things was just that, that big fundraiser that we held. That was a huge thing for us because we informed the community around the project, we had huge support from all over. And not only that, but it really helped other people within the community see … how many people are actually doing this kind of stuff”* (LCEF)

*“I think the collaboration in the group, people are very respectful of each other and outside of the learning circles themselves; we formed a committee and we managed to pull off a fundraiser that raised $[6]000 for student opportunities to go outside of the learning circle and take trips say to Haida Gwaii. I think that’s the main one, the Haida Gwaii trip was the main one. Yea, none of us had ever done a fundraiser before, and thanks to the generosity of the whole community we made $[6]000 in one night with about 100 people attending.”* (Community Interview)

Developing trust between community members and different stakeholders remains a work in progress but the principle of ‘food is medicine’ served as a focus for connection with the development of a number of skills-based initiatives. The improved community connectedness led to a number of successful grant applications that furthered the work of the L.C. 

*“…the history of colonialism means that in this community there’s quite—there’s a difference in the way that Gitksan and non-Gitksan organizations work. There’s a lack of trust. With good reason.”* (LCEF 2017)

The hinge around which LC partnerships and activities swing is the LCEF. However, a heavy reliance on volunteer time and parent education was found to be necessary to keep community and school-based programs running, and a need to build a wider network of support for projects was acknowledged.

In addition to the development of connections within the HZ region, connections between HZ and the LC in HG were developed after annual gatherings, with a visit to HG for some of the students, and a visit from LC participants in HG.

*“It’s an awesome project and I am really glad that we had the opportunity to work with other communities from Saskatchewan and from Haida Gwaii because being isolated and trying to do this on our own would have been a lot. I learned a lot from [both] so that’s been really important … for sure.”* (LC Participant)

*“Yeah, I think it is a powerful working with other nations because there’s a lot of similar challenges and approaches.”* (LCEF)

#### 3.5.3. Focus on Youth: Importance of Healthy Good for Young People

The Indigenous Seventh Generation Principle identifies that decisions made now should support a sustainable future for seven generations hence and speaks to the value placed on youth. This philosophy was a prominent theme in the data. 

*“The children are flowers that need to be cared for and fed. It is our responsibility to work together as a community to care for them so they are valued each as the unique life that they are, so that they are not damaged. Gitksan women are the foundation of raising the next generation of children. Gitksan are a matrilineal people and the relationships on the mother’s side are very important.”* (Community Elder)

Discussions relating to the refocus of the school curriculum towards learning about and celebrating traditional ways took place at the LC meetings, and organizations such as Senden that facilitated hands-on learning for youth were partnered with. In addition, the LC was framed by the worldviews and traditional food practices and knowledge of Gitksan LC participants and others engaged with the initiative.

*“…So yeah, working with the chiefs in Gitwangak, they are just as excited to try new farming techniques and integrate Western colonial systems into their system so that they can take pressure of their hereditary systems and allow them to grow. I’ve been learning a lot, I’ve been really grateful for that perspective. Let’s just use everything that we got and see where we get to”* (Community member)

School food programs such as hot lunch and salad bars were built upon, with the understanding that youth should have access to plentiful and healthy food, and that schools can play a part in food security for the children of the community. This links to food sovereignty.

*“HSS school administration has begun to come up with creative solutions to support students to eat well at school, connect with Senden, and build ongoing community based financial support for healthy food options in the school.”* (LC Report)

School gardens were expanded and/or built, and in the case of two schools, it was reported by communities that these spaces provided a refuge for students who had undergone difficult experiences. Similarly, the pre-existing program at Senden provided a chance for youth experiencing challenges to learn, share their skills with others and heal.

*“Many youth have and are experiencing trauma in their day to day lives—the opportunity to connect while being on the land is vital to supporting them to build resilience and the qualities that will help them address these challenges.”* (LC Report)

Tied into this was the importance of youth being outside and on the land. Land-based learning is important as it encompasses culture, family, food and land. In reports from the LCEF, youth were found to be proud of the work they did outside on the land and valued the effort taken in growing and providing food for themselves, their families, and communities. A sense of purpose was seen to be fostered in youth as a result of these programs, and Elders and older adults involved in the LC considered it a blessing to be able to do work on behalf of the children.

*“I think the most important impact it would have on youth would be just knowing who they are, to the core; having strong grounding in their Indigenous ways of knowing, … and I think when we see our youth struggle…it’s just not knowing—it’s struggling with who you are and kinda floundering around and floating out there; that all of our cultural traditions and ways of knowing are really grounding for us, and we’re really comforted in our own shoes, so it’s important. I think it’s a big piece—it’s a big piece of who we are, and it grounds us in who we are, and it grounds us to the land around us. That’s, uh, the strongest grounding that we can possibly have.”* (LC Partner 2018)

#### 3.5.4. Learning from One Another: Knowledge Sharing

A number of interviewees spoke of an ‘increase in knowledge’ gained by community members during the course of the initiative. ‘Learning’ was mentioned regularly in reports, interviews and other documents, whether it was learning about nutrition and choice, or practical hands-on learning at workshops and farm visits. 

*“…there are things happening. There are lots of really good successes, lots of really good involvement from the schools…. It’s good, we are off to a really good start and hopefully this is something that will keep going and develop into the wider sort of food bank. Not the salvation army food bank, but the idea of having a bank of local plants that grow well in this area. … Just trying to really get local, because climatically we have things that are really well adapted to the area.”* (Community Member)

Stakeholders discovered that they were learning from each other and acknowledged that everyone had knowledge and experience to share, even the younger members of community. 

*“We did a photo voice project with our Senden Agricultural Resource Centre youth, and they’re a little bit older—a lot of them are high school students or in their early twenties—and the way that they view local food, and especially traditional foods, is quite phenomenal. Their knowledge of the culture and the protocol and the amount of respect that’s needed around traditional foods is huge, and they’re quite passionate… about it.”* (LCEF)

Learning was not just limited to issues of food and nutrition, but a number of community members spoke about what they had learned personally about the history of the Indigenous people in the region, and how challenges within the community were frequently framed. 

*“And so, I just felt a lot of personal growth and that’s why I feel this work is really important”* (Research Team Member)

Participants felt that the provincial school curriculum could be updated, in part to increase the value around local foods, and in part to encourage openness and knowledge sharing about the history of Indigenous people in the area; the opportunities to link curriculum-based activities to food and garden activities were seen to be endless.

#### 3.5.5. Local Food

‘Local’ food was considered an inherent part of the initiative from the beginning, with the term being built into the project name. In HZ there were challenges with accessing an affordable supply of local food for school food programs as it was found to be much more expensive than non-local food. 

Funding for food-to-school programs is unreliable and often does not support local food procurement. A further concern about this is that it is difficult to think about local food when poverty and food insecurity remains an issue for many families in the community. 

*“…how hard it is to think about local food when there are so many families with limited access to any food and kids are coming to school hungry. A [local] parent was very concerned that there are not enough resources in school as is and wondered about how we would afford local food.”* (LC Participant)

Relating to procurement, local food in this region is seasonal. The development of a directory to connect children and teachers to knowledge keepers and farmers helped with procurement of local foods for school food programs and community events. 

The concept of ‘shifting value’ towards local foods was built into the goals of the LC with the aim of encouraging the community as part of the learning process to make choices for local food in preference to non-local. 

*“I’m hoping to see a more robust food system here in terms of how everyone is able to access food, I’m really inspired by how much traditional and local food there is, but also the other side of the coin we have you know supermarkets that are full of garbage and imported stuff that we could pretty easily replace”* (LC Participant)

Using local meat and fish in schools was difficult, as legislation relating to food safety prevented these foods being served in schools. In addition, value gaps were identified around food security and local and healthy food. For example, the local high school gives $50 monthly gift cards for students to spend at the Gitanmaax Market for lunch, but the money often does not last a whole month and the students frequently buy energy drinks and unhealthy food with the money. 

## 4. Discussion

The LC:LHF2S initiative is adaptable, and incorporates some of the best practice recommendations for community-based initiatives in Indigenous communities [34,35,36] engaging the entire community in planning and implementation of local and traditional food activities. An LC uses an engaged scholarship or action research approach to engaging community in activity while collecting evidence, and the results of this study, and others [17,37], indicate that LCs are practicable in Indigenous communities, can facilitate building of partnerships and contribute to increased access to local and traditional food among school-aged youth.

While the HZ region had some local food activities prior to 2016, there had been no organized community-wide commitment to the food culture and environment and so the LC initiative in this community focused on partnership development, gardens, community-wide skills work, and youth activities, with a specific emphasis on schools. Traditional foods, knowledge and practices were prioritized in youth-based programs.

Despite the words ‘local food’ being included in the project title, incorporating *traditional* food into the diets and lifestyles of community youth was a major focus of the initiative in this community. Familiarity with traditional knowledge and skills among Indigenous youth has decreased over the past 30 to 40 years in Indigenous communities [38,39,40]. Older Indigenous community members in Upper Skeena were concerned about the low levels of traditional food intake among their youth and the resulting impacts on their long-term health; this is in line with other research [2,41,42,43]. The importance of a holistic view of health, culture and connecting with the land for Indigenous communities has been widely documented [44] and is demonstrated by Indigenous wellness wheels and other comparable wellness models [44,45,46]. The inclusion of traditional foods in the diet and developing associated knowledge and skills have been shown to contribute to food security in other studies [47,48]. Additionally, food and nutrition security, and the need for further focus on food sovereignty has been identified as a concern by community members in Upper Skeena. Comments relating to the need for the inclusion of traditional ways of knowing into the school curriculum echo similar themes in other studies [41].

Despite research indicating that Indigenous communities value school-based nutrition programs [49], the focus on schools in Upper Skeena appeared to be a barrier to the success of the LC. None of the on-reserve schools became involved with the LC, and there were challenges at times with engaging the wider Indigenous community. The legacy of the residential schools [50,51] (and connected with this, gardening) in this community is strong, and using the medium of a western-style school as the focus of this initiative may have limited engagement of some Indigenous community members. In addition, while there was engagement and support from the GGC, leadership of the LC remained with Storytellers’ throughout the course of the initiative, which Storytellers’ staff themselves acknowledged was not ideal. It is impossible to understate the importance of Indigenous leadership in these initiatives; trust: a critical component of relationship building, takes time, especially for ideas coming from outside of community.

However, ‘food is medicine’ and food proved to be a source of connection and a means for the development of trust in this community [52]. The ‘Back to the Land’ program based at the high school aimed to focus on traditional activities such as berry picking and other forms of traditional harvesting, as did the program as Senden. School gardening activities had an acknowledged positive impact on the mental health of the youth, and inclusion of the Gitxsan language in youth programs and the following of local traditional protocols became more important as the initiative progressed. Creative events such as the mid-winter gala contributed to the development of community relationships, with the added benefits of raising money, showcasing the work of the LC and the youth, raising the profile of producers of local food among their consumers, and had as its outcome, culturally respectful learnings for school youth. These factors all combined to help build engagement and trust in Upper Skeena [8,9,10,11]. 

Knowledge exchange became an important feature of the initiative. The cultural importance of sharing of traditional knowledge from Elders, and other knowledge keepers, with youth was significant, and from a research perspective acts as an extended version of successful mentoring programs as reported elsewhere [53]. A number of non-Indigenous community members discussed their own personal growth as a result of learning about the past and understanding more about reconciliation [54]. The journey of the LC in Upper Skeena reflects this—at many times in the process work had to be done to develop relationship and overcome a lack of trust between the two groups that co-exist in the community, before food work could be addressed [14,27].

Finding resources, beyond the project funding, for implementing the initiative within schools was difficult. Food safety legislation relating to the consumption of hunted meats in schools was also a challenge, although steps taken by the exemplar community in Haida Gwaii [17] provided a template for other communities to follow. Added to this, there was a challenge with trying to source reliable funding that would provide food for school programs. Local food was frequently found to be unaffordable within the school budget despite the efforts made to build partnerships with local farmers and growers in order to support the local economy, and program implementation funding was not allowed within this research funding. The LC at HG has illustrated that these issues require time, effort and partnership development beyond the scope of this project. 

Looking to the future, the results from this study highlight a number of things that could be considered when thinking about local or traditional food work in predominantly Indigenous communities. Firstly, thought should be given to the funding timescale. The local food to school process has been described as ‘a tree whose leaves are ‘food procurement’ but the roots are ‘relationship’, which must come first’. Three years is a short time to see progress in a newly begun CBPR project; a significant proportion of time was spent building community engagement and developing relationships and such progress is difficult to measure. The location and context of the community contributes to this time pressure; a short growing season influences local food capacity. As well as the funding timescale, the amount of money available for such projects should be considered: without implementation funding within the research budget, there is a need to source other money for infrastructure and food. Sources of adequate, sustained funding for food related projects remain hard to come by. 

It is beyond the scope of this study to examine long-term outcomes in Upper Skeena. Some participants sensed that some relationships and initiatives would continue while others might be more dependent on external funding and a champion.

### Study Strengths and Limitations

This study has several strengths. The community was given full control over the direction of the initiative; the researchers provided guidance and support as necessary and assisted the community for evaluation. Two people independently coded the interviews, and the descriptive quality of the written reports received from the community was excellent. The community established or enhanced a number of relationships and impressive activities to enhance access to healthy and traditional foods and food skills for youth and support for their holistic wellbeing. 

This study also has a few limitations. While more detailed records about workshops, classroom activities, food produced by the school gardens and food consumed by the children in school would have added depth to the case study, community members perceived the initiatives to be successful. Additionally, it was not possible to gather face-to-face to review the initial themes or consider a more detailed analysis of interview data because of the COVID-19 pandemic. The team depended, instead, on virtual interactions made possible through the relationships developed during the initiative, to discuss the information that was shared.

## 5. Conclusions

In conclusion, this novel LC initiative is a feasible and appropriate way of engaging community and promoting local and traditional foods, knowledge, and practices among Indigenous youth in rural and remote locations similar to the Upper Skeena region, British Columbia, Canada. For similar initiative to be successful in other communities, Indigenous leadership is essential, along with adequate time and funding. In HZ, multiple stakeholders came together to forge connections and develop unique initiatives (such as high school placements via Senden and the community fundraising gala celebrating local foods). Such examples should serve as an inspiration for other communities who are interested in developing a similar program for their youth.

Concerns for future research include the promotion of traditional foods and food skills within school food programs focused on Indigenous youth. The development of Sources of adequate, sustained funding for food related projects also be considered. Implications of Learning Circle activity might be felt at the institutional level (e.g., a school food policy that favors access to affordable local and traditional foods or land-based learning opportunities); a regional level (e.g., a school board policy that supports gardening at every school) or more broadly (e.g., guidelines that access to school nutrition program support should show evidence of broad-based community consultation).

## Figures and Tables

**Figure 1 ijerph-19-15878-f001:**
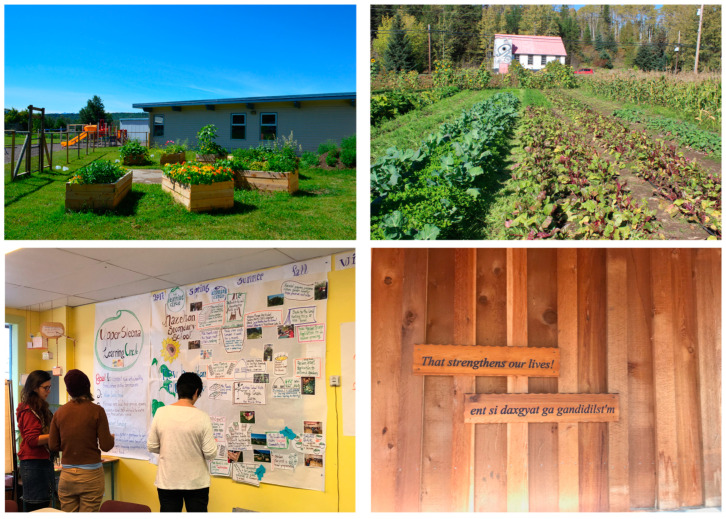
Images from Hazelton/Upper Skeena. Clockwise from top left: MGA school garden; Senden Farm; Sign outside HSS smokehouse; Developing goals during a LC in 2017.

**Table 1 ijerph-19-15878-t001:** Data sources and documents for this study.

Source Category	Total Number of Sources/Respondents
Interviewees/Informants	12
Interviews	18
Photographs	9
Photovoice documents	15
Activity tracking reports	13
LC reports	5
Meeting minutes	15

**Table 2 ijerph-19-15878-t002:** Partners in the LC in Hazelton/Upper Skeena, and associated activities.

Organisation	Focus of Work	Activities
Gitksan Government Commission	Focused on increasing awareness of local, traditional food in the community and its link to wellness via: - Youth - Food security - Partnership development	- Connecting youth: ‘Brighter Futures’ programme that organised youth trips onto the land for berry and medicine picking. - Food security: organised Community Christmas Hampers filled with produce from community gardens. - Partnership development: Provided support to all (Indigenous) communities to enable them to have greenhouses, gardens and smokehouses; organised ‘Wellness Food Festival’ in Winter 2017/8.
Skeena Watershed Conservation Coalition	Food work focusing on: - Food security - Youth - Developing skills - Gardens	- Food security work with communities of Gitanyow and Gitwangak: development of the Gitwangak Food Security Partnership where $8000 of organic produce was delivered to the community of Gitwangak. - Skills and connecting youth through the ‘Youth on Water’ (YOW) river rafting programme: teaching skills and connecting youth to river and culture. - Gardens: development of community gardens in Sik-e-dakh and Gitwanyow communities.
Storyteller’s Foundation	A variety of food activities focusing on: - Gardens - Food security - Youth - Supporting local partner organisations	- Gardens: focus on community and school gardens. - Food security: Kids Get Food programme. - Support for Senden (see below), and MGA and HSS school gardens. - Connecting youth: Facilitating the inclusion of local food into the YOW river rafting programme (SWCC); also running Youth Works Social Programme, a supported youth employment programme that offers catering in the local community
Hazelton Secondary School	A variety of food activities focusing on: - Gardens - Developing skills - Partnership development - Traditional events	- Gardens: Tower gardens (for growing traditional foods), school gardens; building of a root cellar; smokehouse; vermicomposter; planting of a heart garden to assist with grief - Skills development through the ‘Back to the Land’ programme. Foods classes go to Senden Agricultural Resource Centre once a week and workshops with local Elders were held to learn about traditional foods. Filleting/butchering lessons with local fish and moose took place. - Partnership development: Senden, MGA Elementary School (see below); Visit to Haida Gwaii - Indigenous People’s Celebration Day - Tower gardens and vermicomposter purchased with a Farm to School BC grant
Majagaleehl Gali Aks School (MGA)	A variety of food activities focusing on: - Gardens - Developing food skills - Food security - Partnership development	- Gardens: building of garden boxes and greenhouse - Skills: Gardening skills developed in school gardens plus community gardens (‘Tomato challenge’, etc.). Berry picking and food preservation/dehydration workshops - Food security: school food programmes such as the salad bar - Partnership development: Senden Agricultural Resource Centre, Hazelton Secondary School, Wood Grain Farm; Esther’s Chickens - Curriculum activities: Art projects with food - Grants: Farm to Cafeteria Canada for salad bars and school gardens; Mazon Canada for local food procurement
Wood Grain Farm	Agriculture	- Providing local food via the Farmer’s Market; Milling grain; saving seeds
New Hazelton Elementary School	A variety of food activities focusing on: - Gardens - Developing food skills - Partnership development	- Gardens: Garden beds - Skills: Seed planting workshop and field trips to farm and greenhouse (harvesting and seed saving skills); soup-making classes. Chicks were purchased, incubated and hatched. - Partnership development: Storytellers’ Foundation; Wood Grain Farm; Sky High Green House; Senden Agricultural Resource Centre - Grants: Whole Kids Foundation grant for garden expansion project; Received funds from Mid-Winter Gala for cultural visit to Senden Agricultural Resource Centre
Gitanmaax Nursery School	Incorporated local and traditional foods into their curriculum	- Gardens: Flower and vegetable gardens and involvement in the community garden. - Incorporating local and wild foods in children’s snacks
Senden Sustainable Agricultural Resource Centre	Agricultural activities focusing on: - Gardens - Youth - Developing skills - Partnership development	- Gardens: Community-Supported Agriculture box programme - Connecting youth: land-based education for youth. Gitxsenimx language incorporated into programme - Skills: beekeeping; traditional food processing workshops (fish smoking and canning); medicine workshop with Elders - Partnership development: Hazelton Secondary School; MGA bought greens for salad bar programme
Northwest Food Action Network	Developing skills	- Skills: ‘Better Together’ conference (2018)
Dancing Bee Farm	Agriculture	- Providing local food via the Farmer’s Market

## Data Availability

Data are available upon request, providing permission is granted by the requisite community/communities.

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
