# Peer review of "Learning Circles: A Collaborative Approach to Enhance Local, Healthy and Traditional Foods for Youth in the Northerly Community of Hazelton/Upper Skeena, British Columbia, Canada"

_ijerph, 2022, doi:10.3390/ijerph192315878_

Round 1

Reviewer 1 Report

This study described the context, process and outcomes of the Learning Circle (LC) implementation among one of the Indigenous communities in Canada.  The paper needs proofreading. Some sentences are too long with an appropriate writing style. Some references are too old and should be replaced. The abstract, and introduction parts could be shorter. The discussion section must be improved.

Author Response

Reviewer 1:

This study described the context, process and outcomes of the Learning Circle (LC) implementation among one of the Indigenous communities in Canada.  The paper needs proofreading. Some sentences are too long with an appropriate writing style. Some references are too old and should be replaced. The abstract, and introduction parts could be shorter. The discussion section must be improved.

Author’s response:  Many thanks for your comments.  The paper has been re-read and some longer sentences have been shortened.  Grammar and style has been checked throughout.  U.S. spelling has been substituted for U.K. spelling (see for example, analysed to analyzed; programme to program).  Updated references have been added where possible, and the abstract has been reduced.

Reviewer 2 Report

This study investigated the context, process and outcomes of the learning circle implementation in one of the communities, Hazelton/Upper Skeena, located midway between Prince Rupert and Prince George in northern British Columbia. Overall, this study addresses a topic of high relevance for research and also for practice. However, I believe some issues need revision and clarification. 

Reviewer 3 Report

The manuscript entitled "Learning Circles: A collaborative approach to enhance local, healthy and traditional foods for youth in the Northerly Community of Hazelton/Upper Skeena, British Columbia, Canada", describe in an in-depth way the social dimension of healthy and traditional foods within young people in the Northern Comunity  of Hazelton. Although the manuscript is well completed and organised in its various sections, it is very descriptive and, the absence of a re-elaboration of the data and results gives the work the qualities of a report rather than a research article. 

In these regards there are some punctual changes required to be made:
Lines 38-40: You should not introduce the work by mentioning a participant's answer. It would be better to introduce it by highlighting the importance of a collaborative approach in local communities, also by referencing the research with other studies in literature. 

Lines 87-92: Please provide a better and more clear in-depth description of the study purpose.

Line 97: Is the number of 20 participants significant for a community-based participatory research? If yes, please cite adequate references.

The paragraphs 2.5. Data Analysis and 2.6. Context, could be merged into one called "Data Acquisition".

The paragraph 3.3. should be called "Goal definition"

In the results section, the constant presence of participants' own considerable mentions should be elaborated and translated into statistically significant data.

Based on the results re-elaboration, the discussion should be better implemented in accordance with literature studies.

The study strengths is a little bit redonant and unremarkable. It should be implemented.

In general, the article is lack of scientific soundness. Therefore, I suggest major revisions before the acceptance for publication.

Author Response

Reviewer 3

The manuscript entitled "Learning Circles: A collaborative approach to enhance local, healthy and traditional foods for youth in the Northerly Community of Hazelton/Upper Skeena, British Columbia, Canada", describe in an in-depth way the social dimension of healthy and traditional foods within young people in the Northern Comunity  of Hazelton. Although the manuscript is well completed and organised in its various sections, it is very descriptive and, the absence of a re-elaboration of the data and results gives the work the qualities of a report rather than a research article.

Author response:  Many thanks for your kind comments.  We have addressed your concerns below.

In these regards there are some punctual changes required to be made:
Lines 38-40: You should not introduce the work by mentioning a participant's answer. It would be better to introduce it by highlighting the importance of a collaborative approach in local communities, also by referencing the research with other studies in literature.

Author’s response:  We have removed the quote from lines 38-40

Lines 87-92: Please provide a better and more clear in-depth description of the study purpose.

Author’s response:  The study purpose has been adjusted (lines 167-172)

Line 97: Is the number of 20 participants significant for a community-based participatory research? If yes, please cite adequate references.

Author’s response:  Including 20 participants in a qualitative research study exceeds the number included in many other comparable studies (i.e., those  among Canadian Indigenous communities).  Please see the following refs: [2-5]

[2]     Hanemaayer, R.; Anderson, K.; Haines, J.; Lickers, K.R.; Lickers Xavier, A.; Gordon, K.; Tait Neufeld, H. Exploring the perceptions of and experiences with traditional foods among first nations female youth: a participatory photovoice study. International Journal of Environmental Research and Public Health 2020, 17, 2214.

[3]     Skinner, K.; Hanning, R.M.; Metatawabin, J.; Tsuji, L.J. Implementation of a community greenhouse in a remote, sub-Arctic First Nations community in Ontario, Canada: a descriptive case study. Rural and remote health 2014, 14, [79]-[96].

[4]     Skinner, K.; Hanning, R.; Tsuji, L. Barriers and supports for healthy eating and physical activity for First Nation youths in northern Canada. Int. J. Circumpolar Health 2006, 65, 148-161.

[5]     Neufeld, H.T. Food perceptions and concerns of aboriginal women coping with gestational diabetes in Winnipeg, Manitoba. Journal of nutrition education and behavior 2011, 43, 482-491.

The paragraphs 2.5. Data Analysis and 2.6. Context, could be merged into one called "Data Acquisition".

Author’s response:  Many thanks for your suggestion.  We feel, however, that it is important for the reader to understand the unique context within which our study takes place and would like to keep the two sections separate.

The paragraph 3.3. should be called "Goal definition"

Author’s response: Thank you for your suggestion.  We have amended the title of section 3.3.

In the results section, the constant presence of participants' own considerable mentions should be elaborated and translated into statistically significant data.

Author’s response:  We don’t feel that it would be appropriate for us to present our qualitative data in another fashion.  Our current structure adheres to norms for presenting qualitative research which usually aims to provide depth and colour, and the ‘why’ behind the ‘what’, rather than presenting statistically significant results that are generalisable across populations.

Based on the results re-elaboration, the discussion should be better implemented in accordance with literature studies.

Author’s response:  Many thanks for your suggestion.  Due to the novelty of our initiative there are not many other relevant studies with which to compare our work, beyond those that we have already cited. 

The study strengths is a little bit redonant and unremarkable. It should be implemented.

In general, the article is lack of scientific soundness. Therefore, I suggest major revisions before the acceptance for publication.

Author’s response:  Work in Indigenous communities in Canada recognizes the unique context of each Nation,  spanning geography, governance, culture, food systems and more. Qualitative research is well suited to capture this diversity. Moreover, the Learning Circle process we describe is, we believe, remarkable in that it can support positive change in diverse contexts, including the current case study and others referenced from our broader project. We have adhered to principles of rigour in qualitative research methodology. The reviewer is referred to papers that have already been published in this special edition that used qualitative research methods with defined samples of Indigenous participants:

Leclerc A-M, Boulanger M, Miquelon P, Rivard M-C. First Nations Peoples’ Eating and Physical Activity Behaviors in Urban Areas: A Mixed-Methods Approach. International Journal of Environmental Research and Public Health. 2022; 19(16):10390. https://doi.org/10.3390/ijerph191610390

Loukes KA, Anderson S, Beardy J, Rondeau MC, Robidoux MA. Wapekeka’s COVID-19 Response: A Local Response to a Global Pandemic. International Journal of Environmental Research and Public Health. 2022; 19(18):11562. https://doi.org/10.3390/ijerph191811562

Round 2

Reviewer 1 Report

This paper has the potential to be accepted ...

Author Response

No response required.

Reviewer 2 Report

The authors have addressed my comments carefully and my major remarks from the previous version are addressed sufficiently. However, there are some minor points that need to be addressed. 

Reviewer 3 Report

Suggestions were fullfilled in the revised version of the manuscript 

Author Response

No response required.